# Dramatic Differences between the Structural Susceptibility of the S1 Pre- and S2 Postfusion States of the SARS-CoV-2 Spike Protein to External Electric Fields Revealed by Molecular Dynamics Simulations

**DOI:** 10.3390/v15122405

**Published:** 2023-12-11

**Authors:** Alexander Lipskij, Claudia Arbeitman, Pablo Rojas, Pedro Ojeda-May, Martin E. Garcia

**Affiliations:** 1Theoretical Physics and Center of Interdisciplinary Nanostructure Science and Technology, FB10, Universität Kassel, Heinrich-Plett-Str. 40, 34132 Kassel, Germany; uk022176@student.uni-kassel.de (A.L.); clarbeitman@gmail.com (C.A.); pablo.rojas@uni-kassel.de (P.R.); 2CONICET Consejo Nacional de Investigaciones Científicas y Técnicas, Godoy Cruz 2290, Buenos Aires C1425FQB, Argentina; 3GIBIO-Universidad Tecnológica Nacional-Facultad Regional Buenos Aires, Medrano 951, Buenos Aires C1179AAQ, Argentina; 4High Performance Computing Center North (HPC2N), Umeå University, S-90187 Umeå, Sweden; pedro.ojeda-may@umu.se

**Keywords:** SARS-CoV-2, spike protein, structural stability, molecular dynamics simulations, electric fields

## Abstract

In its prefusion state, the SARS-CoV-2 spike protein (similarly to other class I viral fusion proteins) is metastable, which is considered to be an important feature for optimizing or regulating its functions. After the binding process of its S1 subunit (S1) with ACE2, the spike protein (S) undergoes a dramatic conformational change where S1 splits from the S2 subunit, which then penetrates the membrane of the host cell, promoting the fusion of the viral and cell membranes. This results in the infection of the host cell. In a previous work, we showed—using large-scale molecular dynamics simulations—that the application of external electric fields (EFs) induces drastic changes and damage in the receptor-binding domain (RBD) of the wild-type spike protein, as well of the Alpha, Beta, and Gamma variants, leaving a structure which cannot be recognized anymore by ACE2. In this work, we first extend the study to the Delta and Omicron variants and confirm the high sensitivity and extreme vulnerability of the RBD of the prefusion state of S to moderate EF (as weak as 10^4^ V/m), but, more importantly, we also show that, in contrast, the S2 subunit of the postfusion state of the spike protein does not suffer structural damage even if electric field intensities four orders of magnitude higher are applied. These results provide a solid scientific basis to confirm the connection between the prefusion-state metastability of the SARS-CoV-2 spike protein and its susceptibility to be damaged by EF. After the virus docks to the ACE2 receptor, the stable and robust postfusion conformation develops, which exhibits a similar resistance to EF (damage threshold higher than 10^8^ V/m) like most globular proteins.

## 1. Introduction

The SARS-CoV-2 virus continues to be a significant global health concern. Since its emergence in 2019, SARS-CoV-2 has mutated and spread widely among the human population, leading to the emergence of several variants of concern (VOCs) [1]. Among the many monitored variants (Alpha, Beta, Gamma, Epsilon, Lota, Kappa, Mu, Zeta), three have had a significant worldwide impact: Alpha, Delta, and Omicron [2,3]. The latter is particularly noteworthy due to its distinctive feature of having approximately 30 or more mutations in its surface glycoprotein, in sharp contrast to the Delta variant, which has only a few mutations [3]. While the developed vaccines offer protection, some VOCs can evade immunity, highlighting the need for ongoing transmission control measures.

During the process of viral entry, enveloped viruses such as SARS-CoV-2 depend on the fusion of their lipid-enveloped structure with the cell membranes of host organisms. In the case of coronaviruses, this crucial fusion event is driven by the spike (S) protein, a class I viral transmembrane fusion protein that exhibits distinctive features [4,5]. The S protein on the mature virion consists of trimeric polypeptide chains with glycosylated residues on the surface. Each monomeric unit contains both S1 and S2 subunits, which remain noncovalently bonded until viral fusion is initiated [6]. The S1 fragment, comprising the N-terminal domain (NTD), and the receptor-binding domain (RBD), plays a key role in recognizing and binding to the host receptor angiotensin-converting enzyme 2 (ACE2) [7]. The S2 subunit mediates viral cell membrane fusion and mainly comprises two regions known as heptad repeats (HR1 and HR2). These heptad repeats consist of repetitive heptapeptides characterized by α-helical structures and various hydrophobic residues, which are involved in the transition from the prefusion state to the postfusion conformation [8]. Various regions contribute to the modulation of the fusogenic structural rearrangements of the S protein. The ACE2 engagement by the S protein exposes the S2′ site from S2, whose cleavage allows for the release the fusion peptide (FP) domain, an event pivotal for fusion pore formation and which fully activates the fusion process [9,10]. Within the S1 subunit, the departure of the loop situated in subdomain 2 (SD2) from its hydrophobic surface destabilizes this domain, liberating the N-terminal segment of S2 from S1, leading to the release of S1 at the S1–S2 boundary. The subsequent dissociation of S1 sets off a sequence of refolding events in the metastable prefusion S2, facilitating the fusogenic transition to a stable postfusion structure. The FP, situated in the S2 subunit, is introduced into the host cell membrane, while the C-terminus remains anchored within the viral envelope [9,10].

The free energy required for viral membrane fusion to overcome kinetic barriers is derived from the energy released during a substantial conformational shift in the viral envelope S protein [11]. Similar to other class I viral fusion proteins, the S protein typically resides in a prefusion conformation, trapped in a high-energy, metastable state [12,13,14,15]. Upon interaction with the host cell, a remarkable structural rearrangement of the S protein towards a lower energy, stable postfusion state occurs. This process involves the sequential folding of HR2 onto HR1, forming a structure called a six-helix bundle or 6HB in an antiparallel configuration at the fusion core [16]. As a result, the viral membrane is drawn toward the host cell membrane and firmly adheres to it, ultimately allowing for the fusion of the two membranes. The energy barrier restraining the prefusion state has been observed to be particularly low in the case of coronaviruses’ S proteins [17,18].

Both theoretical predictions and experimental evidence have indicated that the application of electric fields (EFs) can induce significant conformational changes in proteins [19,20,21,22]. This phenomenon mainly arises from the balance between conformational and electrostatic energies, along with entropic contributions [23,24]. In a previous study, employing molecular dynamics (MD) simulations, we demonstrated that low to moderate electric fields with intensities as low as 10^5^ V/m can affect the wild type (WT), Alpha, Beta, and Gamma RBDs of the S protein in such a way that they can overcome a non-thermal energy barrier, therefore shifting it to a state exhibiting a conformation between the prefusion and postfusion states [25]. The purpose of this study is twofold. On the one side, we extend our analysis to the Delta and Omicron variants of the RBD of the SARS-CoV-2 S protein and show that, remarkably, EF intensities as low as 10^4^ V/m are enough to destabilize the RBD structure of these two variants and to induce irreversible changes on a sub-microsecond timescale. On the other hand, and as a central result of this paper, we assess the impact of external EFs on the conformational stability of the postfusion S2 non-functional form, taking into account its dynamic structural changes. Interestingly, we observed no significant structural alterations, even when applying high-intensity EFs (10^8^ V/m), which is a threshold for significant conformational changes reported for many globular proteins.

## 2. Materials and Methods

### 2.1. Simulation Setup

The initial coordinates of the protein chains were obtained from the Protein Data Bank, with IDs 6VSB, 7V8B, 7WBP, and 7COT, which were captured at 3.46, 3.20, 3.00, and 2.16 Å resolution, respectively [26]. The simulation cell consisting of the protein chains, waters, and ions was set up with the CHARMM-GUI interface [27,28,29]. The protein chains were centered in cubic cell boxes of sizes 198 nm for postfusion and 106 nm for delta/omicron and then solvated with TIP3P waters [30] and Na^+^ and Cl^-^ ions at 150 mM concentration.

The dynamical propagation of the Newton equations was achieved with the GROMACS simulation package solver (v. 2021) [31,32,33]. A cutoff distance of 12 Å was employed for solving the short-range interactions in both electrostatic terms, with the particle mesh Ewald (PME) method [34], and van der Waals terms. In the former, a fourth order of cubic interpolation method and a grid size of 1.2 Å was used. Hydrogen atoms were constrained with the LINCS algorithm [35].

The protocol for minimization and equilibration of the initial structure was similar to the one in our previous published work [25].

### 2.2. Principal-Component-Analysis

Principal component analysis (PCA) was performed on the trajectories for both EF-on and EF-off states to characterize and visualize these trajectories and their respective states. In order to accomplish this, each trajectory was projected onto a two-dimensional space obtained through dimensionality reduction. Generalized coordinates, specifically dihedral angles, were employed to define the structural states, effectively separating the internal protein motion from its overall motion [36]. Furthermore, each dihedral angle was partitioned into two metric coordinates, signifying its sine and cosine components, respectively. This transition from dihedral space to a linear metric space introduced a well-defined Euclidean distance, ensuring a unique representation while circumventing artifacts resulting from the periodic nature of angles [37].

Subsequently, PCA was applied to the aforementioned metric coordinates to derive a reduced space. The first two components, corresponding to the highest eigenvalues, were selected to define the reduced two-dimensional space. PCA, as a technique, was employed to identify correlated motion patterns by diagonalizing the covariance matrix, where the eigenvectors delineated the directions of collective motion, and the eigenvalues, ranked in descending order, quantified their respective amplitudes.

PCA was executed using the scikit-learn Python library [38]. Following this, each trajectory was projected onto the reduced PCA space. It is important to note that while we employed components associated with trajectories under an electric field (EF) intensity of 10^7^ V/m for projection, similar plots could be generated using principal components corresponding to runs with different EF intensities. Therefore, the conclusions drawn from this analysis are not dependent on the specific choice of EF intensity.

### 2.3. Free-Energy Landscape Estimation

The estimation of free energy was carried out by previously recording the root mean square displacements (RMSD) $\left\langle r^{2}\right\rangle$ of the system. For this purpose, we utilized a path-sampling method [39,40,41] to approximate the potential of mean force (PMF) for each condition (no-EF, EF-on, and EF-off) and for each strength of the electric field. Considering the RMSD as the x-axis to characterize the states of the protein, we determined the free energy profile using the equation:(1)$$F(\langle r^{2}\rangle )=-k_{B}\;T\;\ln(\langle \delta ({\left\langle r^{2}\right\rangle }_{j}-\langle r^{2}\rangle )\rangle )$$where ${\left\langle r^{2}\right\rangle }_{j}$ represents the segmented RMSD value of the *j* position along the trajectory, *k_B_* stands for the Boltzmann constant, *T* is the temperature, and δ (...) denotes the Dirac delta function. Each trajectory was discretized into 20 windows along the RMSD coordinate. This discretization was selected by considering the trade-off of minimizing the error while reconstructing the distribution of the sample population.

### 2.4. MD Analysis

The trajectory files were processed and analyzed using GROMACS tools or the MDAnalysis Python library [42]. The MD trajectories were visualized, and molecular representations were drawn with the assistance of the VMD software package version 1.9.4a38 (2019) [43].

Customized Python scripts and the MDAnalysis library were employed to calculate various residues and atomic distances, including those related to the crucial amino acids. To analyze interactions by determining the distance between individual residues, the mean distance between all the atoms of each residue was computed.

The STRIDE algorithm, integrated into the VMD software package version 1.9.4a38 (2019), was employed to estimate changes in the secondary structure of the RBD over time under the conditions of no-EF, EF-on, and EF-off. The STRIDE algorithm relies on hydrogen bond energy and statistically derived backbone torsion angle data to characterize secondary structures within trajectories previously generated by GROMACS.

### 2.5. Electrostatic Potential Surface Calculations

The calculation of all potential maps on the PDB 7V8B and 7WBP structural data and final frames from the MD trajectories was carried out using the Adaptive Poisson–Boltzmann Solver (APBS) algorithm [44]. The PDB formats were initially prepared using the PDB2PQR web server and converted to PQR format using the CHARMM force field, with PROPKA set at pH = 7.0 [45]. Subsequently, the APBS analysis was conducted via the linearized Poisson–Boltzmann equation in the VMD software, with settings parameters including a solvent dielectric constant of 78.5, a solvent radius of 1.4 Å, a solute dielectric constant of 2.0, a system temperature of 300 K, a surface density of 10.0 points/Å, and the use of harmonic average smoothing for surface definition.

## 3. Results

The results of our simulations are shown in Figure 1, Figure 2, Figure 3, Figure 4 and Figure 5. 

### 3.1. Global Structural Changes in the S1 and S2 Subunits of the Spike Protein under Electric Fields

The impact of external electric fields (EFs) on the overall structure of the S1 and S2 subunits of the S protein was investigated through molecular dynamics simulations. We studied the S1 subunit with a selected segment of the S protein in an “up” conformation, spanning from residue 319 to residue 686 in the prefusion conformation. This segment encompasses the entire receptor-binding domain (RBD), subdomains SD1 and SD2, as well as the interface connecting S1 and S2 (as illustrated in the upper panel of Figure 5). In the simulations without applied EFs, this segment displayed the local structure, dynamic properties, and biochemical attributes consistent with those exhibited in the complete protein, therefore supporting the choice of a segment for numerical modeling [46]. In this manner, simulations were conducted within a confined spatial domain without compromising the generalizability of the findings (also refer to the Methods section). Regarding the S2 subunit, the primary focus was placed on the postfusion core directly using the asymmetric unit as deposited in the Protein Data Bank (PDB) without applying symmetry operations to obtain the 6HB [47]. The simulations were initiated using structures from the PDB (PDB ID 6VSB for the S1 subunit and 7COT for the S2 subunit), and missing residues were added (see Methods). The first production run aimed to attain thermal equilibrium for the system in the absence of an EF (referred to as the “no-EF” run), ensuring that the protein reached thermodynamic equilibrium at 30 °C. The conformation obtained from the experimentally derived segment closely resembled a stable equilibrium folding state, and the thermalization run primarily served to relax any residual structural tension. Subsequently, using the thermally equilibrated structure as the initial state, simulations were conducted on each subunit of the S protein in the presence of an electric field for a duration of 700 nanoseconds (“EF-on” runs). For the study of the prefusion S1 subunit, low field intensities were applied ($10^{4}-10^{7}$ V/m), based on our previous results [25]. For the stable postfusion S2 subunit, we performed simulations assuming an EF intensity of 10^8^ V/m, which is just below the damage threshold for globular proteins. In the case of the prefusion subunit (PDB: 6VSB), protein elongation was observed in the trajectories due to the alignment of permanent local dipoles and the displacement of charges parallel to the electric field (as observed in Figure 1a), also including a loss of tertiary protein structures within a few nanoseconds. The conformational shifts of the prefusion S1 subunit protein under the influence of an electric field are clearly visible in the time-dependent changes in the root-mean-square displacements (RMSD) of the protein backbone relative to the initial structures (Figure 1a). The transition from the initial conformation to a new stable structure occurs within the first 200 nanoseconds.

**Figure 1 viruses-15-02405-f001:**
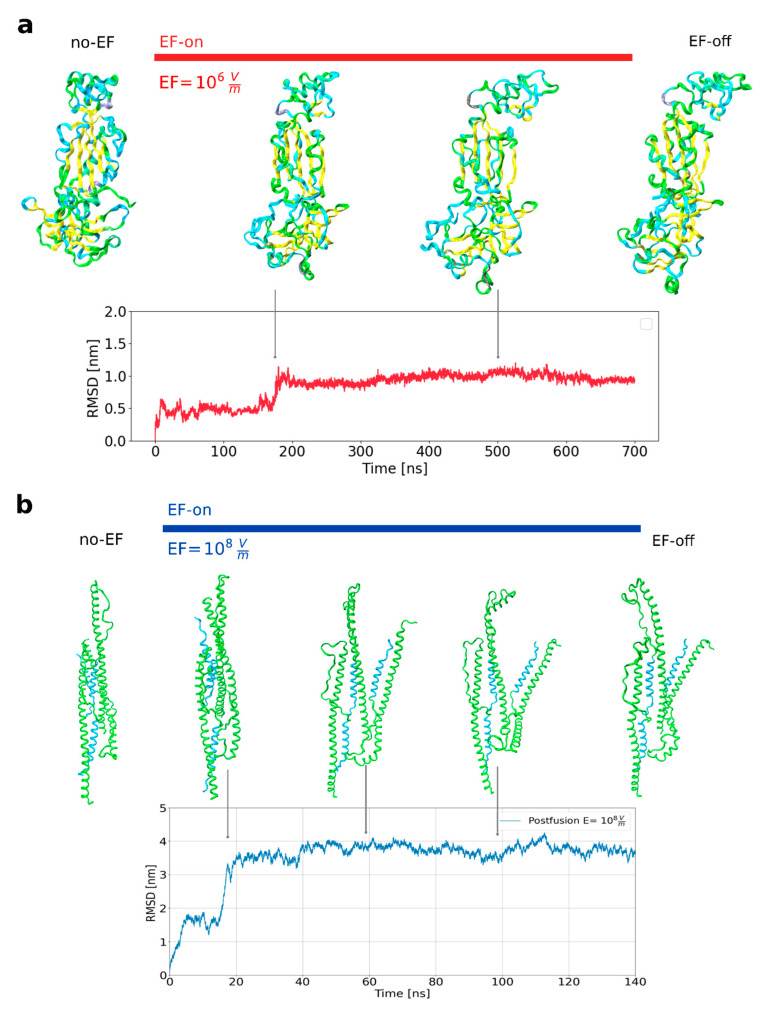
Large amplitude conformational changes are observable in the studied segments of the SARS-CoV-2 spike protein. (**a**) Snapshots of the prefusion conformation (S1 subunit) as it evolves with EF application (upper row). Deviations from the initial structure are quantified with the root mean square deviation (RMSD) (lower row). (**b**) Snapshots of the postfusion conformation (S2 subunit, PDB 7COT) under an EF (upper row) and deviations from the initial structure as RMSD (lower row).

To facilitate a direct comparison of stability between the prefusion and postfusion states, a 140-nanosecond simulation was conducted while applying a high electric field to the postfusion structure (PDB:7COT). The postfusion state of the S protein, under a field strength of 10^8^ V/m, exhibits significantly greater stability in its secondary structure compared to the metastable prefusion state. The overall three-dimensional structure underwent changes during the initial 20 nanoseconds of the dynamics, while maintaining the interaction between the two heptad repeat (HR) regions in each protomer. The simulation starts from an asymmetric unit within which the three HR1-linker-HR2 chains are related by a three-fold axis forming a homotrimeric complex. A few seconds after applying the EF, as a result of conformational rearrangements, the single chains move away from each other and each molecule acts as a protomer. As can be seen in the RMSD (Figure 1b), this conformation is maintained throughout the entire simulation without significant additional change. This global dynamic has its explanation in the presence of flexible loops that connect HR1 and CH in the prefusion state, which are preserved in the asymmetric 7COT unit along with the flexible linker that connect HR1 and HR2. These results could have functional implications in terms of avoiding the formation of the HR1-HR2 six-helix bundle, critical for viral entry mediated by class I fusion proteins [48].

### 3.2. Effects of Moderate Electric Fields on the Secondary Structure of the Receptor Binding Domain

The RBD has a unique three-dimensional structure that contains critical amino acid residues relevant to the virus specificity and binding affinity to the ACE2 receptor. The receptor binding motif (RBM) is the primary functional component within the RBD which forms the interface responsible for the interaction between the spike protein and the ACE2. Mutations located distally from the binding region have been identified as factors affecting the structural stability of the prefusion spike protein and its affinity to ACE2, highlighting the important role played by the precise spatial arrangement of RBM residues involved in receptor binding. Loop 3 (L3) encompasses amino acids from Tyr470 to Pro491 and is one of the four loops constituting the RBM. L3 is crucial in S protein-ACE2 interaction, due to the presence of critical β-strands (Cys488-Tyr489 and Tyr473-Gln474) that significantly enhance WT SARS-CoV-2 affinity for ACE2. This enhanced affinity, estimated to be approximately 15–20 times greater than that of SARS-CoV-1, results from the structured β-strands within L3, which stand in contrast to the unstructured L3 in SARS-CoV-1 [49]. In our earlier work, the influence of EF on the stability of the RBD and its essential residues involved in the local interaction with ACE2 was examined for the wild-type RBD and several VOCs. The current objective is centered on the extension and generalization of results to the impact of EFs on the ability of S to dock into the ACE receptors. To achieve this, a series of simulations were conducted to determine whether the Delta and Omicron VOCs are also susceptible to damage caused by EFs of low to moderate strength. MD simulations were conducted running over 700 ns for every variant, using a model based on the structure of the unbound RBD, which was derived from experimental data acquired from the PDB structure of the RBD-ACE2 complex (PDB 7V8B and 7WBP).

Initially, for both Delta (PDB:7V8B) and Omicron (PDB:7WBP) variants of the RBD, thermalization simulations without an electric field (EF = 0 V/m, “no-EF”) were conducted, and it was observed that the secondary and tertiary structure were preserved in comparison to the original crystal structure. Throughout the thermalization simulations, the mentioned β-strands in the L3 loop remained intact, in agreement with prior studies that emphasize the rigidity of β-sheets in WT SARS-CoV-2, contributing to the stability of L3. Concerning the Delta and Omicron variants, they present mutated residues located in the RBM region. While the Delta variant has two mutated residues, Leu452Arg and Thr478Lys, Omicron exhibits 15 substitutions throughout the RBD, which are expected to cause a change in the spatial organization of RBD residues as well as RBD-ACE2 interactions [50,51]. Nevertheless, previous research has emphasized the importance of L3 stability in the ACE2-interacting interface of the RBD [48]. Furthermore, the overall secondary structure of the RBD was closely monitored and found to exhibit remarkable stability throughout the thermalization run (Figure 2a,b,d,e).

**Figure 2 viruses-15-02405-f002:**
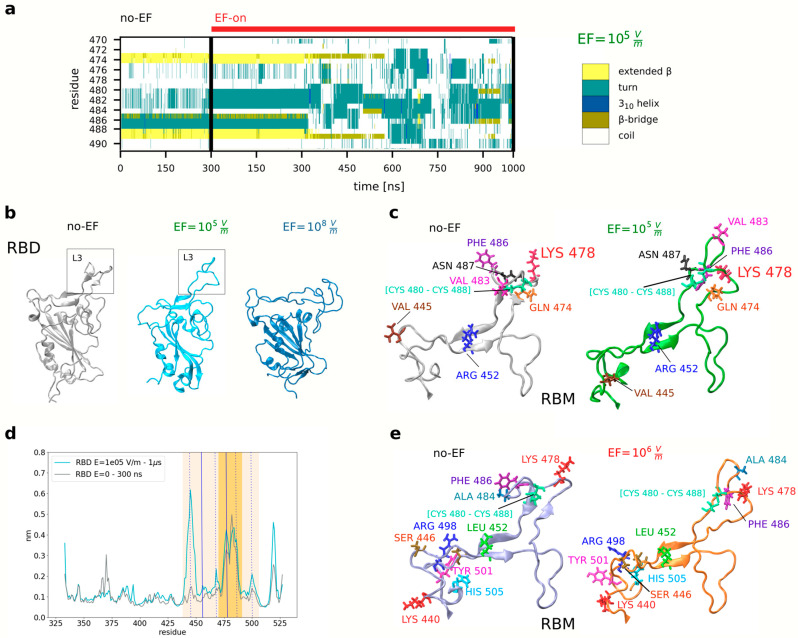
EFs of different moderate intensities induce changes in the relative positions and orientations of residues in the RBD spike protein of the Delta and Omicron variants of concern of SARS-CoV-2. (**a**) Evolution of the secondary structure in the residues corresponding to the loop L3 of the RBM of the prefusion structure of the Delta variant under an EF (PDB 7V8B). The beta-sheets (yellow) disappear completely after 600 ns. (**b**) The relative position and structure of the residues in the loop L3 (Delta variant) are changed after applying 10^5^ V/m and 10^8^ V/m, underlying the removal of the beta-sheet structure (middle and right panels) and random coil formation at 10^8^ V/m (right panel). (**c**) Repositioning of residues at 10^5^ V/m (right panel). (**d**) Root mean square fluctuations of the RBD reveal flexibilization of the backbone structure under application of EF. (**e**) Close-up view showing the key residues of the RBD that participate in the binding with ACE2 for the Omicron variant, both before and after EF application.

To investigate the impact of EFs, simulations were carried out at different field intensities: EF = 10^4^, 10^5^, 10^6^, and 10^7^ V/m. Figure 2a shows the time evolution analysis of L3 secondary structure (for the Delta variant, EF = 10^5^ V/m) obtained using the STRIDE algorithm implemented in VMD [43] on the simulation files. A gradual reduction in the β-sheets was observed during the initial 300 ns of the EF-on simulation, resulting in their eventual deconstruction into turns or random coils before 1 μs. A representative snapshot of the no-EF and EF-on runs (an example for EF = 10^5^ V/m) for the Delta variant shows that the secondary structure of the RBD experienced disruptions at multiple segments, with a notable impact on L3 (Figure 2b). As a result, L3 goes through a transition from its compact structure with the two β-sheets to an open and entirely unstructured coil, similar to the conformation of L3 in SARS-CoV-1 [49,52]. In addition to the previous analysis, the impact of EF on the flexibility of the RBD was also investigated, considering the distinct flexibility properties of coil or loop structures compared to highly ordered secondary structures like β-sheets and α-helices in proteins. To quantify these changes, root-mean-square fluctuations (RMSF) analysis of the RBD, which describes the flexibility of the residues, was employed. In Figure 2d, it is demonstrated that the flexibility of the RBD is modified by the EF in a non-uniform manner, with a more impact on the L3 loop and the RBM region.

For the sake of comparison, we also performed simulations (of 130 nanoseconds duration) of the impact of high field strengths (10^8^ V/m) on the RBD of the Delta variant. In Figure 2b (right panel) the snapshot of the conformation after 60 ns. The disruption on two neighboring β-strands that forming a small two-stranded antiparallel β-sheet in the RBM (in addition to the L3) into an open unstructured coil can already be observed. Comparing this simulation to the previous ones conducted under lower field strengths, the destruction and alteration of the secondary structure is stronger and occurs at a much faster rate, within the first 60 ns.

The same type of simulations and evaluation to assess the impact of EFs was performed on the Omicron variant, where it was observed that the disruption of beta sheets is an irreversible process, and L3 turned into an open and unstructured state (Figure 2d). These findings agree with our prior research on the wild-type and other S protein variants [25]. Taken together, these observations strongly imply that the initial conformation of the RBD is destabilized by the application of an EF. As a result, the secondary structure of crucial RBM segments involved in the docking to ACE2 undergoes changes, leading to the disruption of the spatial atomic organization of backbone and side chain residues in key positions. These rearrangements have important implications for RBD-ACE2 interactions, which are weakened.

### 3.3. Stability of the Field-Induced Final Conformational States in the RDB Spike Protein

To gain insights into the conformational changes, a PCA was performed on the EF-on trajectories to represent and visualize the protein states in a two-dimensional space obtained through dimensionality reduction. By considering a subspace defined by the two most relevant principal components for the run at EF = 10^7^ V/m, the trajectories for all EF-off runs were projected onto that plane (Figure 3a, Delta variant). It was observed that under the influence of EFs with different intensities, different paths in the phase space were explored by the protein. The final conformation at the end of each EF-on run was found to depend on the specific intensity of the applied field, suggesting field-dependent directions of movement in the high-dimensional space of internal coordinates. Once the external field was deactivated, during the EF-off runs, the protein structure remained in the vicinity of the EF-induced new conformations, and there was no return to the initial structures. Figure 3a, shows the points corresponding to the trajectories clustered around the final states, with almost no overlapping regions in the reduced phase space, in agreement with previous results [25].

**Figure 3 viruses-15-02405-f003:**
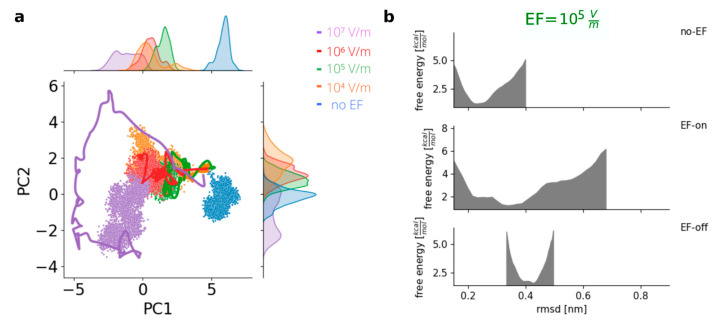
Irreversible states are achieved in the Delta variant of concern after the application of an EF. (**a**) Principal component analysis shows distinct stable states after the application of an EF (lines: smoothed trajectory during EF, dots: states visited with EF-off). (**b**) Energy profile estimation of the prefusion conformation (PDB 7V8B) before the application of an EF (no-EF), during the application of an EF (EF-on) and after the application (EF-off); example for EF = 10^5^ V/m.

To investigate the stability of the new, EF-induced conformations of the S protein, the EF was deactivated, and the simulations were continued in the absence of fields (“EF-off” run). On average, no more than 200 ns were required for each EF-off run, as the protein exhibited limited motion around the structure formed after the EF application for all EF intensities. The existence of a new stable minimum for all studied EF intensities was confirmed by the estimated free energy plots, in line with previously published research, and the presence of an energy barrier preventing a return to the original conformation was also revealed. In Figure 3b, the middle panel shows both the initial state and the damaged state of the system, both under the influence of the EF. The free energy profile connecting both states in the absence of fields was also determined, clearly showing a barrier separating the two states. An observation worth noting at this point is that the minimum state achieved after applying the external field is not a transient or rapidly decaying metastate. Instead, it proves to be remarkably stable and long-lasting.

### 3.4. Disruption of the Charge Complementarity between RBD and ACE2 upon the Application of an Electric Field

Electrostatic complementarity in protein–protein interactions plays a crucial role in molecular recognition and binding processes, as more than 20% of all amino acids in proteins become ionized under physiological conditions, and polar groups are present in sidechains [53]. The results presented in the previous sections demonstrate that the changes in the secondary structure and atomic rearrangement in the S protein, induced by the EF, are likely to weaken its interaction with ACE2.

To investigate whether the EF-induced atomic reorganization affects the electrostatic potential landscape of the RBD, potentially leading to a reduced interaction with the ACE2 receptor, we conducted an in-depth analysis of the electrostatic potential (φ) within the receptor binding motif (RBM). To calculate φ, we utilized the Adaptive Poisson–Boltzmann Solver (APBS) algorithm, which allowed us to solve the Poisson–Boltzmann equations for continuum electrostatics. Before the analysis, we prepared the PDB formats using the PDB2PQR web server, converting them to PQR format with the CHARMM force field, and adjusted PROPKA to pH = 7.057. Subsequently, we performed the APBS analysis using the Linearized Poisson–Boltzmann equation within the VMD software (see Section 2).

Figure 4 shows a significant distortion in the spatial distribution of φ on the RBM when an EF of intensity 10^6^ V/m is applied. This distortion is present in both the Delta and Omicron variants. Since mutations in SARS-CoV-2 variants lead to different interactions at the RBD:ACE2 interface, it is important to take into account the specific ACE2 binding surface for each variant. The mutations at the RBD interface induce significant perturbations in the van der Waals and electrostatic interactions for both the Delta and Omicron variants [54,55]. In the S protein of the Delta variant, the ACE2 binding surface exhibits two prominent positive patches attributed to residues Lys 31 and Lys 353, which precisely match with the corresponding negative regions on the RBD (Tyr 489, Gln 498, and Thr 500) [51]. The negative areas within the structure are distinguished by the presence of polar and acidic residues, namely Gln 24, Tyr 83, Asp 355, and Gln 493. These residues contribute to a stronger electrostatic complementarity at the binding interface with Lys 417, Asn 487, and Lys 478 [54], as illustrated in Figure 4. Notably, specific mutations, such as Thr 478 to Lys and Leu 452 to Arg, play a crucial role in altering the electrostatic surface of the RBM. These mutations involve replacing uncharged amino acids with positively charged ones [54]. This modification in the electrostatic properties of the RBM is significant as it can impact the interactions and binding affinity with the receptor site. The introduction of positively charged residues through mutations may enhance the binding capabilities of the viral protein to its target, influencing the overall infectivity or recognition by host cells. During the thermalization process (no-EF run), no significant changes were observed in this region. However, upon the rearrangement of L3 resulting from the EF application, a shift in the negative region within the RBD is observed. Specifically, positively charged residues, primarily Arg 452, shift and match the positive part of ACE2, resulting in the generation of a repulsive force. Comparing the electrostatic landscape before and after applying the electric field highlights discernible alterations, ultimately resulting in a diminished binding affinity. This reduction is attributed to the essential requirement for charge complementarity between the RBD and ACE2. The alteration in the potential surface property is attributed to the rearrangement of residues within the RBM, as depicted in both figures. Another contributing factor is the reorientation of polar groups in both the solute and solvent, influenced by the presence of electric fields and local charges.

**Figure 4 viruses-15-02405-f004:**
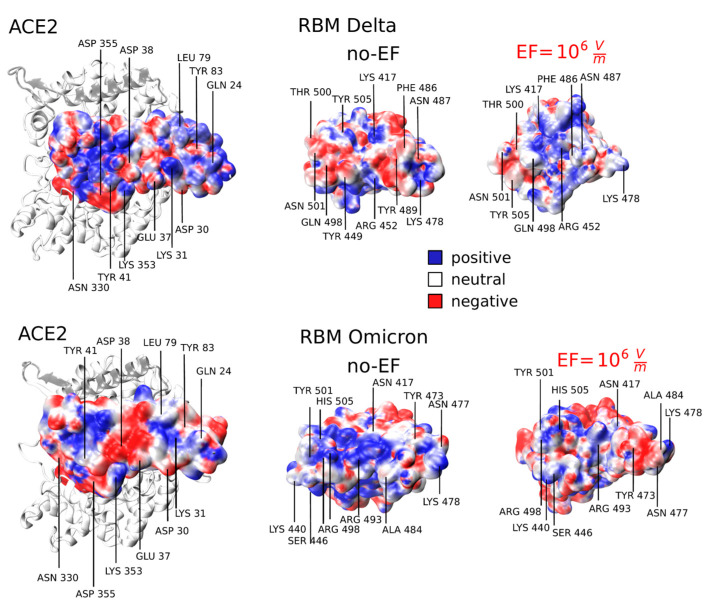
Electrostatic potential at ACE2 and the receptor binding motif (RBM) interface under EF conditions (EF = 0 and EF = 10^6^ V/m). Red, white, and blue colors represent negative, neutral, and positive charges, respectively (0.3 eV/−0.4 eV). It visualizes the disruption of charge complementarity between RBD and ACE2 upon EF application.

In the case of the Omicron variant, and as shown in Figure 4, mutations entirely alter its interaction profile. The considerable number mutations, 15 in the RBD, leads to an increase in the positive electrostatic potential in the RBM when compared to the WT and other variants, enhancing the affinity for interactions with the large negative (or neutral) regions of the ACE2 receptor [55]. The size of the positively charged regions in Omicron is significantly larger compared to Delta and WT, explaining why the electrostatic attractive forces are stronger in the Omicron RBD-ACE2 interaction compared to WT and other VOCs [55]. Upon application of an EF, we can observe a disruption in the central positive patch which results in a repulsive contact with the ACE2 surface. These findings on the electrostatic properties indicate that the surface charge distribution in the RBM is significantly altered by the EF, disrupting the electrostatic complementarity between the S protein and ACE2. Consequently, we can argue that EF leads to a potential disruption in the bonding between the RBD and ACE2, resulting in a reduction in their electrostatic affinity.

### 3.5. Influence of Electric Fields on the Fusion Core Region of the S2 Subunit of the Spike Protein

In the postfusion state, numerous strong interactions exist between the HR1 and HR2 domains within the helical section identified as the fusion core. The HR1 domains form a trimeric coiled-coil core that runs in parallel, surrounded by three HR2 domains in an antiparallel fashion [56]. We demonstrated above that EF application induces conformational changes in the tertiary structure of the postfusion S2 subunit, possibly destabilizing the 6HB structure. To better understand the structural effects of EF on the secondary structure in the fusion core, we analyzed HR1-HR2 interactions represented in single-chain mode. As shown in Figure 5, relevant interacting residues of HR1 that contribute to binding with HR2 through the formation of hydrogen bonds and salt bridges were conserved [48].

These results suggest that under an EF, the secondary structure is not affected in the postfusion S2 subunit. However, the conservation of interactions between the HR1 and HR2 domains does not necessarily imply further stabilization of the 6-HB structure, which may lead to increased virus infectivity [57].

The fact that, according to our simulations, the postfusion conformation does not suffer irreversible damage in its secondary structure upon the application of electric fields as strong as 10^8^ V/m is consistent with the widely accepted damage threshold between 5 × 10^8^ V/m and 10^9^ V/m for globular proteins, which was determined by different theoretical and experimental works. Notice also that the alpha helices are much more robust and resistant to the application of electric fields, as has been analyzed in detail in Ref. [20]. It is therefore not surprising that the postfusion conformation is mainly composed of alpha helices, since this structure must withstand the electric field through the cell membrane, which typically has an intensity around 10^9^ V/m.

**Figure 5 viruses-15-02405-f005:**
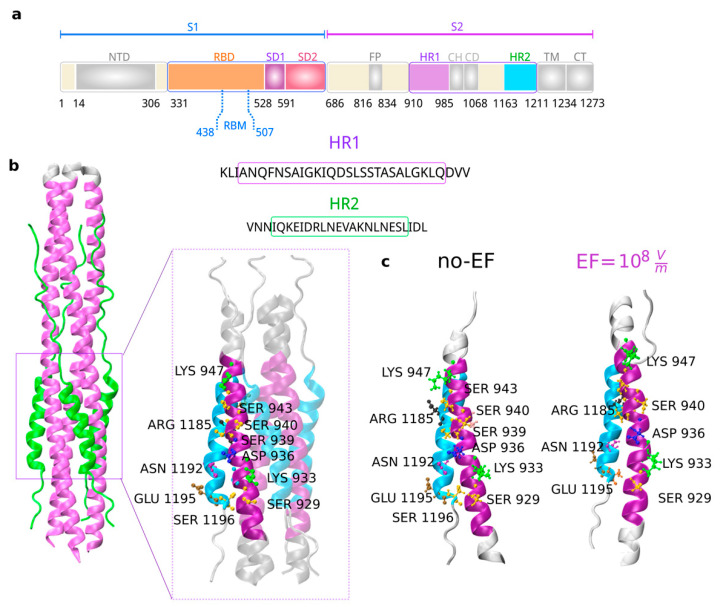
(**a**–**c**) Schematic representation of the S1 and S2 subunits within the SARS-CoV-2 spike protein, along with a sequence alignment of the HR1 and HR2 domains. The segments used in this study have been distinctly highlighted using colors (upper panel). The core of the S2 subunit in the postfusion state (PDB 6LXT) and structural details of the HR1 and HR2 region interactions are presented in cartoon form, with HR1 highlighted in purple and HR2 in cyan (PDB 7COT). Important residues are indicated and labeled. The fusion core regions for each protomer remain conserved after the application of an EF.

## 4. Discussion

In this work, we studied the effects of external electric fields on the conformation of the SARS-CoV-2 spike protein using molecular dynamics simulations. We focused on the prefusion and postfusion conformations in order to characterize the differences in the stability and the vulnerability of both structures.

The different conformational states of the spike protein used in this study to assess the effect of applying external electric fields to the protein were selected based on a previously proposed model [58]. In the model, cryo-electron microscopy structures for both the prefusion and postfusion states of the S protein are extracted and analyzed to conclude that the spontaneous transition to the postfusion state is independent of target cells. It is suggested that both prefusion and postfusion states are present on the surface of the mature virion in variable proportions [59,60]. We remark that the starting hypothesis and main focus of our manuscript consist in the application of an EF directly to the S protein or possibly to the isolated virion, with the aim of producing severe conformational changes that then affect recognition or binding to the receptor. Thus, since the central objective of applying electric fields is to produce viral inactivation, we evaluated the effect of such EFs on both the prefusion structure (represented by the S1 segment in our study) and the postfusion structure corresponding to subunit 2 (once the S1-ACE2 complex dissociation from subunit 2 has occurred). The S1 is already previously cleaved from the S2, only held together by non-covalent interactions. The separation of S1 is the end result of conformational changes after binding to ACE2. This postfusion structure differs from the conformation that subunit 1 in the prefusion state has, mainly at the NTD, NTD-associated subdomain, and RBD-associated subdomain levels that change position. Therefore, it would not be a case of interest to consider subunit 1 in the postfusion state in this study.

Our study shows that the receptor binding domain of the prefusion conformation of the spike protein of the Delta and Omicron variants undergoes significant changes in the tertiary as well as the secondary structure under the application of EFs. In contrast, the postfusion conformation displays changes only in the tertiary structure, which can be attributed to its slender shape that contributes to its flexibility.

We characterized the changes in the structures by combining detailed analysis of the position and orientation of individual residues, and changes in the secondary structure of segments that participate in interaction with ACE2, together with global descriptors of the trajectories that are derived from the calculation of the free energy landscapes, dimensionality reduction and computation of the electrostatic potential in the surface of the protein.

A detailed analysis of the RMSF data revealed an increase in flexibility for L3 (residues 470 to 491) even in absence of fields for Delta variant (Figure 2d), similar findings were also obtained in previous works for other variants when compared to WT [50]. Under EF application, residues of the RBD that considerably increase their flexibility, particularly Val445, Phe456, Ile468, Phe486, and Thr500 (Figure 2d, dashed lines), along with a moderate increase in flexibility for the mutated residues Arg452 and Lys478 (Figure 2d, continuous lines), which are critical and situated at the interface region to the ACE2 receptor. In the Delta variant, some of these residues have significant contributions to ACE2 binding through the formation of hydrogen bonds or salt bridges [61]. It is worth mentioning that, although Arg452 and Lys478 may not directly interact through bonds with ACE2, previous studies have highlighted their influence on the structural microenvironment of crucial interface residues, such as Gln493 and L3 [51,62,63]. A similar scenario occurs in the Omicron variant, where some of the mutations do not directly participate in the docking to ACE2 but may affect the microenvironment of interacting residues.

The analysis of the trajectories in the conformational space displayed by the proteins, which we performed via PCA and the calculation of the free energy landscapes along the visited states, reveal that the final states resulting from the application of EFs to the prefusion conformation are stable and clearly distinct from the initial structures. Follow-up work should address the comparison of the dynamics under EFs of the full spike protein trimer in the prefusion and postfusion conformations, in order to identify the complete set of conformational changes and, in consequence, characterize the spatial and residue dependence of the susceptibility to EFs.

Overall, this work provides evidence that the reported vulnerability of the SARS-CoV-2 spike protein resides in its metastable characteristic, which is present in the prefusion but not in the postfusion conformation. Other class I viral fusion proteins are also metastable in their prefusion conformation, a fact that is described to participate in membrane fusion processes. This suggests the possibility that the reported vulnerability to EFs, which is demonstrated here for multiple variants of the SARS-CoV-2 spike protein, extends as a general feature in class I fusion proteins of other viruses.

Regarding the practical applications of these results, a potential avenue for implementation could involve the development of air filtration systems utilizing electric fields of moderate intensities. Such filters could effectively neutralize aerosols by inactivating the RBD of the spike proteins. This not only underscores the significance of this research in understanding the dynamics of viral structures but also opens a field for innovative solutions with tangible benefits in public health and safety.

## Data Availability

The data presented in this study are available on request from the corresponding author.

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
