# Peer review of "Dramatic Differences between the Structural Susceptibility of the S1 Pre- and S2 Postfusion States of the SARS-CoV-2 Spike Protein to External Electric Fields Revealed by Molecular Dynamics Simulations"

_viruses, 2023, doi:10.3390/v15122405_

Round 1

Reviewer 1 Report

Comments and Suggestions for Authors

The authors analyze the effects of external electric fields on the conformation of the Delta and Omicron  variants of SARS CoV2 Spike protein.

Important point

Since the journal is not specific for physicists, it would be appropriate to add the significance of this type of study in the field of biology and what application implications they have.

In the 3.1 results, the authors use an EF intensity of 104-107 Vm-1 for the study of prefusion S1 subunit and an EF intensity of  108 Vm-1 for the study of postfusion S2 subunit.  Why don't they use the same EF intensity?

Minor points

line 182: Protein Data Bank -> Protein Data Bank (PDB)

line 192: prefusion subunit -> prefusion S1 subunit

line 196: add PDB: 6VSB

line 251: specify delta and omicron variants

line 366-370: review and explain better

figure 3b: is it possible use the same scale? Unfortunately using different scales the legend is not on the same line

figure 4: add also wild-type RBD

text and figures: use either Vm-1 or V/m

Reviewer 2 Report

Comments and Suggestions for Authors

The manuscript, following another published paper (ref. 23), is interesting although I find some details that could be improved. First of all, I think the title is misleading. There are differences in the stability of the different parts of the protein but they are not as significant as one might initially think when reading the title of the manuscript. Secondly, and this is my impression, one might think that there is a direct relationship between the pre- and post-fusion states (they are the same entities), but they are two completely independent domains (the comparison is about two completely different subunits of a unique protein). I don't know if I understood it correctly and maybe I'm wrong, but the title should be changed to reflect this fact.

Comments.

.- In the introduction I think it would be necessary to name in a more specific way the different domains involved in the fusion process : the N-terminal fusion peptide, the fusion domain and the internal fusion peptide.

.- What are the structures corresponding to the prefusion form of the S1 subunit?. What are the structures corresponding to the postfusion form of the S1 subunit?. What are the structures corresponding to the prefusion form of the S2 subunit?. What are the structures corresponding to the postfusion form of the S2 subunit?. Clearly define them in the manuscript. And clearly define the structural changes observed for each one of them.

.- Page 3, line 141. Why have 20 windows been used?. Why not less?. Why not more?.

.- Page 5, lines 204-206. Again, one thinks that two different states of the same domain have been studied, but that is not the case, two different domains have been studied. It must be clarified very well what has been done.

.- Figure 1. You present the results for the S1 subunit and for the S2 subunit. For the S1 subunit it is the prefusion conformation. Where is the postfusion conformation? For the S2 subunit it is the postfusion conformation. Where is the prefusion conformation?. You present the data using 10^6 V/m for S1 and 10^8 V/m for S2. Shouldn't they be the same?.

.- It is my opinion that the work should be limited only to the S protein (the title says it all) and its subunits and not mix it with the ACE2 domain. Remove the corresponding section, i.e., 3.4.

.- It seems to me that different field intensities have been used for each of the subunits studied. Shouldn't be used the same intensities so that the structural changes, if any, could be compared directly?

.- The discussion should comment on the results between the pre- and postfusion states of S1 and S2.

Minor comments.

.- Page 3, line 117, and page 8, line 307. PCA has been defined previously.

.- Figure 5. Divide the figure (A, B, ...., etc.). The legend is not very well understood.

Round 2

Reviewer 1 Report

Comments and Suggestions for Authors

The article is much improved. Good work.

Reviewer 2 Report

Comments and Suggestions for Authors

The manuscript could be published as it is now.